# Clinical Evaluation of Metagenomic Next-Generation Sequencing Method for the Diagnosis of Suspected Ascitic Infection in Patients with Liver Cirrhosis in a Clinical Laboratory

Hao-Xin Wu,[a] Fei-Li Wei,[b] Wei Zhang,[a] Jie Han,[a] Shan Guo,[b] Zheng Wang,[a] De-Xi Chen,[b] Wei Hou,[a] Zhong-Jie Hu[a]

[a]Beijing YouAn Hospital, Capital Medical University, Beijing, China
[b]Beijing Institute of Hepatology, Beijing YouAn Hospital, Capital Medical University, Beijing Precision Medicine and Transformation Engineering Technology Research Center of Hepatitis and Liver Cancer, Beijing, China

Hao-Xin Wu and Fei-Li Wei contributed equally to this paper. Author order was determined by contribution to draft writing.

**ABSTRACT** Metagenomic next-generation sequencing (mNGS), mostly carried out in independent clinical laboratories, has been increasingly applied in clinical pathogen diagnosis. We aimed to explore the feasibility of mNGS in clinical laboratories and analyze its potential in the diagnosis of infectious ascites. Two reference panels composed of 12 strains commonly appearing in peritonitis were constructed to evaluate the performance metrics based on in-house mNGS protocols. The mNGS clinical detection value was analyzed in 211 ascitic samples and compared with culture and composite standards. Finally, eight patients with cirrhosis were prospectively enrolled to verify the clinical value of mNGS in peritoneal infection diagnosis. The mNGS analytical performance showed that the assay had great linearity, specificity, stability, interference, and limits of detection of 33 to 828 CFU/mL. The sensitivity and specificity of mNGS for bacterial or fungal detection using culture standards were 84.2% and 82.0%, respectively. After adjustment using digital PCR and clinical judgment, the sensitivity and specificity increased to 87.2% and 90.1%, respectively. Compared with culture, mNGS detected a broad range of pathogens and more polymicrobial infections (49% versus 9%, $P < 0.05$). The pathogen results were obtained within 24 h using mNGS in eight prospective cases, which effectively guided antibiotics therapy. mNGS testing in clinical laboratories affiliated with a hospital has certain advantages. It has unique superiority in pathogens detection, particularly in patients with polymicrobial infections. However, considering spectrum characteristics and test cost, pertinent pathogen panels should be developed in clinical practice.

**IMPORTANCE** This study established and evaluated a complete metagenomics next-generation sequencing assay to improve the diagnosis of suspected ascitic infection in a clinical laboratory affiliated with a hospital. The assay is superior to traditional culture testing and will aid in the early and accurate identification of pathogens, particularly in patients with polymicrobial infections. This assay is also essential for precision therapy and can reduce the incidence of drug resistance stemming from irrational use of antibiotics.

**KEYWORDS** metagenomic next-generation sequencing, ascitic infection, diagnosis, clinical laboratories

Metagenomics next-generation sequencing (mNGS) has emerged as an important method in detecting emerging, rare, coinfectious pathogens owing to its advantages of rapid and unbiased testing, a broad range of pathogen detection, and high accuracy (1, 2). The application of mNGS testing in clinical laboratories is restricted to some degree by complicated operations, skilled technicians, and strict lab environments, among others. Currently, mNGS testing is mostly carried out by the laboratory-developed test model in independent clinical laboratories at home and abroad (3). However, knowledge of mNGS detection has

Address correspondence to Zhong-Jie Hu, huzhongjie@ccmu.edu.cn, or Wei Hou, baoerlanglang@163.com.

The authors declare no conflict of interest.

improved with technology in clinical practice, and the demand to detect pathogens in clinical laboratories affiliated with hospitals has gradually increased (4).

Bacterial infections, particularly spontaneous bacterial peritonitis (SBP), occur in approximately 40% to 70% of patients with cirrhosis and might play an important role in the progression of liver failure, development of liver-related complications, and mortality (5). Bacterascites, defined by an ascitic fluid polymorphonuclear neutrophil (PMN) count below 250 cells/mm$^3$ and positive ascites culture results, may represent the first step in the development of SBP (6). Many recent studies on several diseases, such as meningitis and sepsis, have demonstrated the high diagnostic accuracy of mNGS testing (7, 8), but to our knowledge, there have been few studies on ascites. Previous studies on ascites only enrolled 73 or fewer samples for detection, lacked accuracy, or reported only three case series (9–11), and so a comprehensive study of bacterial characteristics is lacking.

Etiological diagnosis is the most important part of diagnosing infectious diseases, and culture is considered the "gold standard" for pathogen diagnosis. However, practical application is limited given its low sensitivity (6) and long turnaround period (12). Furthermore, the diagnostic value of culture for mixed infection caused by multiple pathogens is poor, and cannot meet clinical needs (13). With the extensive application of molecular biology techniques (14, 15), bacteria identification through PCR technology, such as by using 16S rRNA, has been attempted in clinical practice. However, owing to its limitations of low throughput and poor specificity, it is not widely used in clinical diagnosis and treatment.

Therefore, our study aimed to establish and evaluate a set of mNGS assays in a clinical laboratory affiliated with a hospital. The mNGS assay was conducted to characterize the pathogenic spectrum of infectious ascites and analyze its clinical value and feasibility for infection diagnosis.

## RESULTS

**Patient characteristics.** Among the 205 enrolled patients, three cohorts of patients were analyzed, including 66 SBP cases (32.2%), 37 bacterascites cases (18.0%), and 102 no-AFI (49.8%). The SBP cohort showed increased white blood cell and neutrophil counts in blood and in ascites compared with those in bacterascites and no-AFI groups ($P < 0.001$). The demographic data are presented in Table S2.

**Analytical performance characterization of mNGS testing.** To calculate the minimum concentration at which 95% of microorganisms were detected, the probit analysis was conducted, which showed that the limit of detection of bacteria ranged from 33 to 828 CFU/mL and that of fungi was 579 CFU/mL (Table 1; Table S3). The results showed a good linear relationship between CFU and the nRPTM detected by mNGS, $R^2 = 0.95$ to 0.99 (Fig. S2). The interference results showed that the nRPTM of *Staphylococcus epidermidis* and *Staphylococcus aureus*, and *Enterococcus faecalis* and *Enterococcus faecium* detected by mNGS was 1:9.53 and 1.47:1, respectively, indicating that mNGS could accurately distinguish the species in the same genus (Table 1; Table S4). For precision and stability, the mNGS testing showed no difference in nRPTM at different time points, the intra-assay run and each replicate interassay run. The average coefficient of variation was 16.7% (Table 1; Excel S1).

The receiver operator characteristic (ROC) curve was plotted based on the training set ($n = 67$ samples, 31 positive and 36 negative) relative to the conventional culture standard. The results showed that nRPTM $\geq 10$ maximizes the efficiency of pathogens detection (Fig. S3A).

The sensitivity and specificity of mNGS testing based on the validation set ($n = 138$ samples) compared to traditional culture were 84.2% (95% CI = 72.1% to 96.4%) and 82.0% (95% CI = 74.3% to 89.7%), respectively. The positive predictive value (PPA) and negative predictive value (NPA) were 64.0% (95% CI = 50.2% to 77.8%) and 93.2% (95% CI = 87.8% to 98.6%), respectively (Table 1). There were 18 positive mNGS detections with negative culture results. Furthermore, considering the low sensitivity of culture, a composite standard of ddPCR and clinical adjudication was adopted, which found that six of the 18 cases were positive using ddPCR, and three cases had obvious clinical manifestations (Table S5). After adjustment with the composite standard, the sensitivity and specificity of mNGS detection increased to 87.2% (95% CI = 77.3% to 97.1%) and 90.1% (95% CI = 83.9% to 96.4%),

**TABLE 1** Analytical performance characterization of mNGS testing[a]

| Performance metric | Methods | Results |
|---|---|---|
| Limit of detection (LOD) | Representative microorganism | LOD (nRPTM) |
| | *Enterococcus Faecium* | 114.3 |
| | *Streptococcus agalactiae* | 42.8 |
| | *Escherichia coli* | 353.4 |
| | *Klebsiella pneumoniae* | 46.9 |
| | *Candida glabrata* | 579.4 |
| Linearity | Goodness-of-fit | Results |
| | $R^2$ | 0.95 to 0.99 |
| Interference | Mixture | nRPTM |
| | *Staphylococcus epidermidis* and *Staphylococcus aureus* mixed at 1:10 | 35,925:342,691 (1:9.53) |
| | *Enterococcus faecalis* and *Enterococcus Faecium* mixed at 2:1 | 120,810:81,960 (1.47:1) |
| Precision and stability | Methods | Results |
| | Qualitative detection over 6 consecutive PC runs (intraassay) | 100% concordance |
| | Qualitative detection of five PC samples on the same run (interassay) | 100% concordance |
| | 4°C for 0, 12, 24h | 100% concordance |
| Accuracy | Diagnostic standard | Diagnostic performance |
| | Conventional clinical testing | Sensitivity = 83.8% |
| | | Specificity = 81.2% |
| | | PPA = 62.0% |
| | | NPA = 93.2% |
| | Conventional clinical testing + Digital Droplet PCR + Clinical adjudication | Sensitivity = 87.2% |
| | | Specificity = 90.1% |
| | | PPA = 82.0% |
| | | NPA = 93.2% |

[a]PPA, positive predictive agreement; NPA, negative predictive agreement; nRPTM, normalized reads per 10 million; PC, positive control.

respectively. PPA and NPA were 82.0% (95% CI = 71.0% to 93.0%) and 93.2% (95% CI = 87.8% to 98.6%), respectively (Table 2; Fig. S3B).

**Analytical pathogens characteristics in patients with infected ascites.** The ascitic pathogen analysis of 205 patients showed that the positive rate of culture was 34.6% (71/205), and the positive rate of mNGS detection was 40.5% (83/205), of which Gram-positive bacteria were the main pathogens (Fig. 1A). Compared to the culture, the mNGS showed that the proportion of bacteria mixed infection increased to 34% (28/83, versus 10% [7/71] of culture), with different types of Gram-positive and Gram-negative bacteria. The detection accuracy of mNGS (n = 37; 18 G+, 9 G−; four fungi, five viruses, and one parasite) was higher than that of culture (n = 23; 13 G+, 8 G−; two fungi) in terms of pathogen numbers and spectrum (Fig. 1B).

In patients with advanced cirrhosis ascites, the presence of bacteria does not necessarily cause the development of SBP (16). According to the clinical composite diagnosis, 66 of 103 patients with suspected peritonitis were diagnosed as having SBP, among which the leading pathogens were, for example *E. faecalis*, *E. faecium*, *S. aureus*, *K. pneumoniae*, *E. coli*, and *E. cloacae* (Fig. 1C). These bacteria were also the common pathogens reported in previous ascitic studies (9, 13). However, approximately 46% of the samples were negative, even with mNGS testing. The main bacteria in 37 patients with bacterascites were *S. epidermidis*, *S. haemolyticus*,

**TABLE 2** Ascitic pathogens in patients with spontaneous bacterial peritonitis

| | Culture | | | | mNGS | | | |
|---|---|---|---|---|---|---|---|---|
| Ascitic pathogens | All (n = 33) | Monomicrobial (n = 30) | Polymicrobial[a] (n = 3) | P value | All (n = 35) | Monomicrobial (n = 18) | Polymicrobial[a] (n = 17) | P value |
| Bacteria | 32 (97%) | 29 (97%) | 3 (100%) | 0.748 | 35 (100%) | 18 (100%) | 17 (100%) | 1.000 |
| Gram-positive | 21 (64%) | 19 (63%) | 2 (67%) | 0.909 | 27 (77%) | 14 (78%) | 13 (76%) | 0.927 |
| Gram-negative | 11 (33%) | 9 (30%) | 3 (100%) | 0.016 | 13 (37%) | 4 (22%) | 9 (53%) | 0.022 |
| Fungi | 1 (3%) | 1 (3%) | 0 (0%) | 0.748 | 3 (9%) | 0 (0%) | 3 (18%) | 0.062 |

[a]Polymicrobial infections included mixture of two Gram-positive bacteria/Gram-negative bacteria or Gram-positive with Gram-negative bacteria or bacteria with fungi.

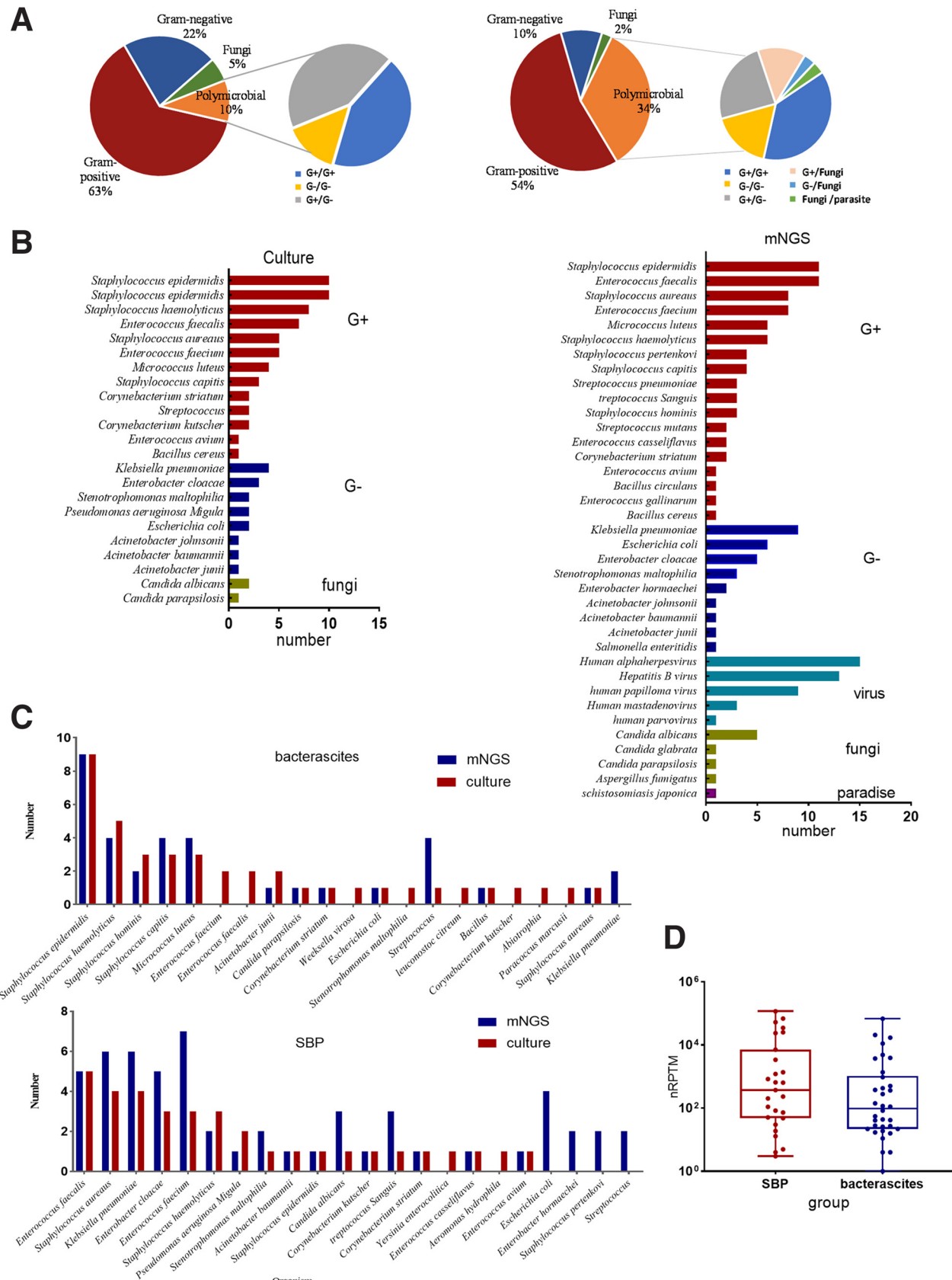

**FIG 1** Comparison of pathogens spectrum between cirrhosis ascites culture and mNGS identification (A). Comparison of culture and mNGS identification in terms of pathogen classification categories (B). Comparison of culture and mNGS identification in terms of pathogen species (C). Pathogenic characteristics that appeared in SBP (*n* = 66) and bacterascites (*n* = 37) (D). Comparison of nRPTM between SBP and bacterascites. nRPTM, normalized reads per 10 million; mNGS, metagenomics next-generation sequencing; SBP, spontaneous bacterial peritonitis.

*Micrococcus luteus*, and *Streptococcus* (Fig. 1C), which were reported to have relatively weak pathogenicity in recent studies (17, 18). Additionally, the nRPTM count showed that the DNA concentration of patients with SBP was higher than that of patients with bacterascites, although the difference was not significant ($n = 0.15$) (Fig. 1D).

Of the 66 patients diagnosed with SBP based on clinical standards, pathogens were detected in 33 patients by culture and in 35 by mNGS. Microbial characterization showed that spontaneous monomicrobial infection occurred in 51% (18/35) and polymicrobial infection occurred in 49% (10 patients with two types of pathogens, six with three types, and one with four types; Fig. S4A) by mNGS. In addition, we found that SBP caused by Gram-negative bacteria may be associated with mixed infection of multiple microorganisms ($P < 0.05$, Chi-square test) (Table 2). Furthermore, the 6-month mortality of polymicrobial infection was higher than that of monomicrobial infection ($P = 0.015$; Fig. S4B).

**Analytic and diagnostic differences between mNGS and conventional culture testing.** Subsequently, we compared the diagnostic value of different microorganisms in 39 patients with SBP between mNGS and culture. For Gram-positive bacteria, the traditional culture showed a sensitivity, specificity, PPV, and NPV of 72.4% (95% CI = 54.3 to 85.3%), 90% (95% CI = 59.6 to 98.2%), 95.5% (95% CI = 78.2 to 99.2%), and 53% (95% CI = 31.0 to 73.8%), respectively. However, the mNGS results showed a sensitivity, specificity, PPV, and NPV of 93.1% (95% CI = 78.0 to 98.1%), 90% (95% CI = 59.6 to 98.2%), 96.4% (95% CI = 82.3 to 99.4%) and 81.8% (95% CI = 52.3 to 94.9%), respectively. For Gram-negative bacteria, the culture results showed a sensitivity of 81.3% (95% CI = 57.0 to 93.4%) and a NPV of 88.5% (95% CI = 71.0 to 96.0%) compared to the 93.8% (95% CI = 71.7 to 98.9%) sensitivity and 95.8% (95% CI = 79.8 to 99.3%) NPV of mNGS testing. Notably, the consistency of the two detection methods for Gram-negative bacteria was higher than that for Gram-positive bacteria (0.71 versus 0.42 by Kappa test), but this difference was not significant ($P > 0.05$ McNemar test; Table 3). For the diagnostic value of different types of pathogens, the accuracy of culture was equal to that of mNGS, even with the improved sensitivity of mNGS detection.

**Comparative analysis in SBP diagnosis with three detection methods.** We compared the diagnostic accuracy rate between culture and mNGS for diagnosing SBP. The results showed that pathogens were detected in 50.0% (33/66) of patients by culture, 54.5% (36/66) of patients by mNGS. Pathogens were detected in four patients by culture only, including *Y. enterocolitica*, *E. faecalis*, and *S. haemolyticus*, and in seven patients by mNGS only, including *E. faecalis*, *S. aureus*, *E. coli*, *S. sanguis*, *M. luteus*, *S. pertenkovi*, and *S. pneumoniae*. Using digital PCR, pathogens were further detected in 69.7% (46/66) of patients (Fig. S5A). Of the 26 patients with both negative culture and mNGS results, pathogens were detected in 15 patients by ddPCR, including three patients with Gram-positive bacterial infection, eight patients with Gram-negative bacterial infection, and four patients with both. In addition, the specificity and PPA of ddPCR for suspected SBP were higher than that of culture and mNGS (87.1% versus 71.9% and 67.6%, 71.9% versus 45.8% and 44.5%, $P < 0.05$; Fig. S5B).

**Case series.** Of eight patients prospectively enrolled in our study, three were diagnosed with SBP based on international standards, and five with PMN count below 250 cells/mm$^3$, but significant clinical symptoms were clinically diagnosed with SBP (Table 4). The mNGS detection in six cases was faster than that of culture, which provided an early etiological basis. Five patients were culture-negative but mNGS-positive, and the mNGS results were further verified by clinical adjudication and the ddPCR method. The culture showed monomicrobial infection in three patients, whereas the mNGS found polymicrobial infection of Gram-positive bacteria combined with fungi and Gram-positive bacteria combined with Gram-negative bacteria. After the timely adjustment of antibiotics according to the results of mNGS, the patients were discharged with improved symptoms (Text S1).

## DISCUSSION

We established a high-throughput sequencing assay for peritoneal infection and performed a complete performance evaluation in a clinical laboratory affiliated with a hospital. The advantages are as follows: (i) the detection accuracy of mNGS was significantly increased, and a broad

**TABLE 3** Diagnostic accuracy of culture and mNGS in SBP

| Clinical pathogens | Culture | | | | mNGS | | | |
|---|---|---|---|---|---|---|---|---|
| | Sensitivity(%) | Specificity(%) | PPV(%) | NPV(%) | Sensitivity(%) | Specificity(%) | PPV(%) | NPV(%) |
| Gram(+) bacteria | 72.4 (54.3 to 85.3) | 90.0 (59.6 to 98.2) | 95.5 (78.2 to 99.2) | 53.0 (31.0 to 73.8) | 93.1 (78.0 to 98.1) | 90.0 (59.6 to 98.2) | 96.4 (82.3 to 99.4) | 81.8 (52.3 to 94.9) |
| Enterococcus faecalis | 66.7 (30.0 to 90.3) | 97.0 (89.3 to 100) | 80.0 (37.6 to 96.4) | 94.1 (80.9 to 98.4) | 83.3 (43.7 to 97.0) | 100 (89.6 to 100) | 100 (56.6 to 100) | 97.1 (85.1 to 99.5) |
| Enterococcus faecium | 50.0 (18.8 to 81.2) | 100 (89.6 to 100) | 100 (43.9 to 100) | 91.7 (78.2 to 97.1) | 100 (61.0 to 100) | 97.0 (84.7 to 99.5) | 85.7 (48.7 to 97.4) | 100 (89.3 to 100) |
| Staphylococcus aureus | 66.7 (30.0 to 90.3) | 100 (89.6 to 100) | 100 (51.0 to 100) | 94.3 (81.4 to 98.4) | 100 (61.0 to 100) | 100 (89.6 to 100) | 100 (61.0 to 100) | 100 (89.6 to 100) |
| Other Gram(+) bacteria | 64.3 (38.8 to 83.7) | 92.0 (75.0 to 97.8) | 81.8 (52.3 to 94.9) | 82.1 (64.4 to 92.1) | 87.5 (64.0 to 96.5) | 91.3 (73.2 to 97.6) | 87.5 (64.0 to 96.5) | 91.3 (73.2 to 97.6) |
| Gram(−) bacteria | 81.3 (57.0 to 93.4) | 100 (85.7 to 100) | 100 (77.2 to 100) | 88.5 (71.0 to 96.0) | 93.8 (71.7 to 98.9) | 100 (85.7 to 100) | 100 (79.6 to 100) | 95.8 (79.8 to 99.3) |
| Escherichia coli | 33.3 (6.1 to 79.2) | 100 (90.4 to 100) | 100 (20.7 to 100) | 94.7 (82.7 to 98.5) | 100 (43.9 to 100) | 97.2 (85.8 to 99.5) | 75.0 (30.0 to 95.4) | 100 (90.1 to 100) |
| Klebsiella pneumoniae | 66.7 (30.0 to 90.3) | 100 (89.6 to 100) | 100 (51.0 to 100) | 94.3 (81.4 to 98.4) | 83.3 (43.7 to 97.0) | 100 (89.6 to 100) | 100 (56.6 to 100) | 97.1 (85.1 to 99.5) |
| Other Gram(−) bacteria | 90.0 (59.6 to 98.2) | 96.6 (82.8 to 99.4) | 90.0 (59.6 to 98.2) | 96.6 (82.8 to 99.4) | 90.0 (59.6 to 98.2) | 100 (88.3 to 100) | 100 (70.1 to 100) | 96.7 (83.3 to 99.4) |
| Fungi | 50.0 (9.4 to 90.6) | 100 (90.5 to 100) | 100 (20.7 to 100) | 97.4 (86.5 to 99.5) | 100 (34.2 to 100) | 97.3 (86.2 to 99.5) | 66.7 (20.8 to 93.9) | 100 (90.4 to 100) |
| Candida albicans | 50.0 (9.4 to 90.6) | 100 (90.5 to 100) | 100 (20.7 to 100) | 97.4 (86.5 to 99.5) | 100 (34.2 to 100) | 97.3 (86.2 to 99.5) | 66.7 (20.8 to 93.9) | 100 (90.4 to 100) |

**TABLE 4** Eight cases of ascites mNGS testing in patients with established or probable infection[a]

| Cases | Presentation | Culture | Digital PCR (copies/$\mu$L) G+ | G− | mNGS | Antibiotic treatment |
|---|---|---|---|---|---|---|
| #1 | Alcoholic; fever; elevated PCT, CRP; PMN (102 cells/mm$^3$) | E. faecalis | Pos | Neg | E. faecalis, E. faecium, C. albicans | Empiric Latamoxef treatment; replaced with Piperacillin/tazobactam and Fluconazole (2 days later), discharged with a better health condition. |
| #2 | Fever; gastrointestinal tract bleeding; abdominal tenderness; Child-Pugh grades C; PMN (27 cells/mm$^3$) | Neg | ND | ND | E. faecium | Cefotaxime sodium/sulbactam treatment, Teicoplanin was added 2 days later. |
| #3 | Viral (HBV+HCV); liver cancer; abdominal pain; elevated PCT, CRP; PMN (8,000 cells/mm$^3$) | S. maltophilia | Neg | Pos | E. cloacae, E. hormaechei, E. coli, S. maltophilia, K. pneumoniae | Vancomycin and imipenem treatment, changed to piperacillin-tazobactam (5 days later); discharged without symptoms of infection. |
| #4 | Liver cancer; fever; abdominal pain and tenderness; elevated PCT; PMN (1,188 cells/mm$^3$) | K. pneumoniae | Pos | Pos | K. pneumoniae, E. faecalis, E. faecium | Empiric latamoxef treatment, changed to imipenem and tigecycline after 2 days. Patient's condition improved. |
| #5 | Alcoholic; fever; abdominal pain and tenderness; acute renal failure; elevated PCT; PMN (241 cells/mm$^3$) | Neg | ND | ND | E. faecalis | Imipenem treatment; discharged with a better health condition. |
| #6 | Primary Biliary Cholangitis; abdominal pain; hematochezia; slight elevated PCT; PMN (44 cells/mm$^3$) | Neg | Neg | Neg | E. coli | Empiric Latamoxef treatment, discharged with a better health condition. |
| #7 | Cryptogenic; gastrointestinal tract bleeding; acute renal failure; PMN (13 cells/mm$^3$) | Neg | Pos | Neg | E. coli | Biapenem treatment; discharged with a better health condition. |
| #8 | Abdominal pain and tenderness; elevated PCT, CRP; PMN (128 cells/mm$^3$) | Neg | ND | ND | E. faecalis | Imipenem treatment; vancomycin was added 5 days later. |

[a]PCT, procalcitonin; CRP, C-reactive protein; PMN, polymorphonuclear; ND, not done; Pos, positive; Neg, negative.

range of pathogens were detected, particularly for polymicrobial infections; and (ii) the pathogenic results were rapidly obtained within 24 h using mNGS assay compared to culture, especially when culture and PCR results could not guide clinical treatment (Fig. S1). The mNGS results provided a basis for further antibiotic use.

There are two types of laboratories used in the mNGS process: wet lab and dry lab. Our mNGS test was based on a joint in-house protocol, that is, an in-house part in the wet lab and a complete automated report interpretation system developed by an independent clinical laboratory. Subsequently, we conducted a detailed performance evaluation and feasibility confirmation of the assay. On the one hand, the protocols greatly reduced detection difficulty in the bioinformatic analysis. On the other hand, it reduced the risk of sample transfer, particularly for some infectious samples, and promoted communication and determination of infections and pathogens for clinicians and laboratory technicians.

We enrolled 205 patients with cirrhosis and ascites to comprehensively characterize the pathogenic spectrum of ascites using mNGS and found that mNGS could simultaneously detect more pathogens than culture (37 versus 23), including bacteria, fungi, viruses, and parasites. Our data showed that the positive rate of Gram-positive bacteria in ascites exceeded 50%, about twice that of Gram-negative, which is similar to a recent study on the detection of pathogens in peritoneal infection (9). In addition, although a previous study reported that SBP is typically a monobacterial infection (13), effective antibiotic coverage of pathogens could be limited when solely based on culture, leading to severe clinical outcomes. Our results showed that mNGS could detect a large number of samples with polymicrobial infections and the 6-

month mortality of those patients significantly increased, which was consistent with recent studies (19–21). There are several possible reasons for this. First, selected the incorrect antibiotics due to polymicrobial infection leads to a delay in treatment in clinical practice. Second, liver-cirrhosis immune dysfunction may predispose the host to more severe infection by translocated bacteria, which contributes to an increased risk of mortality (22). Third, previous research has shown that polymicrobial infection is "polymicrobial biofilm formation" and bacterial coaggregation, in which polymicrobial interactions of different pathogens enhanced the pathogenesis and evolution of peritoneal infection (23).

We categorized the patients into three groups: SBP, bacterascites, and no-AFI based on PMN counts and clinical symptoms. Consistent with a previous study (10), nearly 50% of the samples were negative in SBP patients even with the sensitive mNGS assay, but pathogens were detected in 57% of those negative samples by ddPCR. Recent studies indicated that mNGS detected a broad range of pathogens and is appropriate for diagnosing rare infections and intractable diseases, whereas ddPCR is useful for identifying and excluding common peritoneal pathogens because it is faster (4 h versus 24 h) and more sensitive than mNGS when diagnosing SBP (24, 25). In addition, we found that the common pathogens encountered in clinical practice were present in patients with SBP and the most frequent pathogens in bacterascites were mainly Gram-positive bacteria which had lower bacterial load and poor pathogenicity (Fig. S4D). One of the limitations of molecular diagnostic technology is the high false-positive rate, and the frequent detection of contaminants and colonizing pathogens affects the specificity of both NGS and ddPCR when diagnosing peritoneal infection. We considered that high concentrations of bactDNA are usually associated with true infection, whereas low concentrations are associated with either true infection or commensal/colonizer/contaminant microorganisms with unknown clinical significance (26). Therefore, the concentration and characteristics of bactDNA detected by molecular technology and the entire clinical condition must be taken into account in considering the clinical significance.

From eight prospective patients with peritonitis, we found that mNGS testing not only provided a solid basis for clinical precision medicine but also performed better antibiotic management and further improves clinical outcomes. In addition, the average length of patient stay was 12 to15 days. Recent studies have shown that rapid and accurate identification of pathogens using nucleic acid tests decreases health care costs as well as mortality (8, 27). However, the mNGS method is still relatively expensive compared with traditional culture.

This study had some limitations. First, the diagnostic and therapeutic value of viral infections, such as human herpesvirus, is limited in peritoneal infection. Therefore, we could not verify the viral infection in this study. In addition, we could not evaluate the performance for parasites detection owing to the absence of standard strains. Second, spontaneous fungal peritonitis is rare and less studied, but observational data suggest a worse prognosis (6, 28). In the present day, we did not verify the fungal infection because only 6 patients with fungal infection were detected by mNGS. Therefore, it is necessary to develop specific tests for peritoneal fungal infection in the future.

In conclusion, performing the mNGS assay in a clinical laboratory affiliated with a hospital has obvious advantages. It has unique superiority in pathogen detection, particularly in patients with polymicrobial infections. However, considering the spectrum characteristics and test cost, pertinent pathogen panels should be developed for extensive clinical practice in the future.

## MATERIALS AND METHODS

**Study design and samples collection.** The study design was composed of laboratory and clinical studies. First, the mNGS detection workflow was established in a clinical laboratory, and the performance metrics were evaluated using two panels constructed with 12 strains commonly appearing in peritoneal infection. Additionally, 211 patients with cirrhosis were retrospectively enrolled from the Liver Disease Center, Beijing YouAn Hospital, Capital Medical University between May 2020 and September 2021. Patients with secondary peritonitis and sequencing failure and without ascitic culture simultaneously were excluded. Ascitic samples of the first baseline cohort from 205 patients were obtained. The mNGS was used to investigate the ascites microbiome and analyze the clinical diagnostic value compared with culture and composite standard (digital

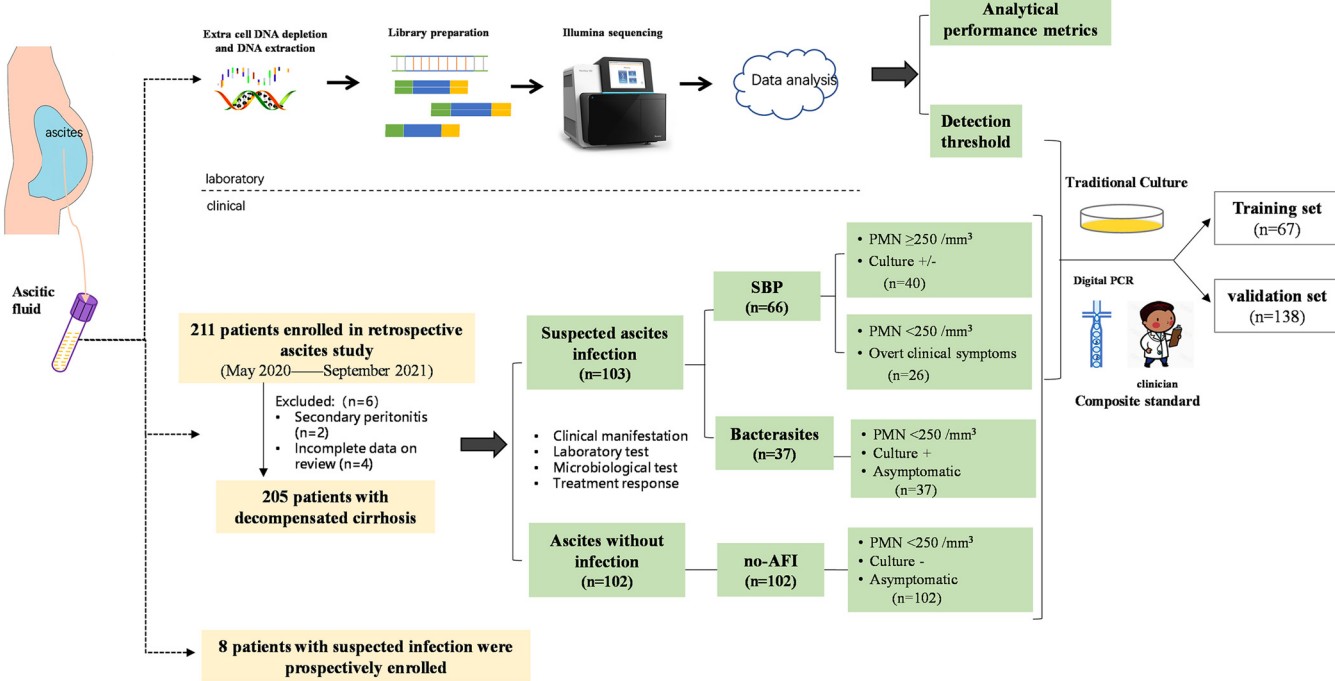

**FIG 2** Study workflow of mNGS detection. Analysis workflow of mNGS testing in the laboratory and for patients with cirrhosis and ascites. SBP (n = 66), bacterascites (n = 37), no-AFI (n = 102); 205 patients were enrolled and divided into the training set (n = 67) for obtaining the threshold and validation set (n = 138) for accuracy analysis. mNGS, metagenomics next-generation sequencing; SBP, spontaneous bacterial peritonitis; AFI, ascitic fluid infection.

droplet PCR and clinical adjudication). Finally, we prospectively enrolled eight patients with suspected or confirmed peritoneal infection. The guiding significance of antibiotics treatment and clinical outcomes was further analyzed (Fig. 2). This study was approved by the ethical committee of Beijing YouAn Hospital, Capital Medical University.

The patients were categorized into three groups: SBP, bacterascites, and ascites without infection (no-AFI). SBP diagnosis was based on PMN and clinical composite diagnosis (2017 Chinese guidelines [12]) that incorporates: (i) clinical manifestation, (ii) laboratory test abnormalities, and (iii) independent adjudication by an infectious disease specialist (C.L.H.) and two liver disease experts (Y.H. and W.H.). The diagnostic standard and inclusion and exclusion criteria are detailed in Text S1 in the supplemental materials.

The ascitic samples were collected at the same time as routine blood cultures that were processed using the BD Bactec FX and BD BACTECTM 9120 blood culture system (Becton, Dickinson and Company) (Text S1).

**Study workflow for mNGS and bioinformatics processing.** The mNGS detection process includes host cell removal, nucleic acid extraction, library preparation, sequencing, and bioinformatics analysis (Fig. 2; Fig. S1). Briefly, DNA was extracted using PathoXtract Universal pathogen enrichment and Extraction Kit (Willingmed) in a 1 mL sample. After DNA purification, 50 $\mu$L DNase/Rnase-free water was added to collect the target DNA. The sequencing libraries were constructed using the Illumina Nextera XT kit (Illumina). Qubit 2.0 (Thermo Fisher Scientific) and Agilent 4200 Bioanalyzer (Agilent Technologies) were used to evaluate the quality of the libraries. Only the libraries with high quality were used for sequencing on the NextSeq CN500 platform (Illumina) using NextSeq 500/550 High Output Kit v2.5 (Illumina) at 75 cycles. Negative control (NC) and positive control (PC) were set for each sequencing run. The protocols are detailed in Text S1 in the Supplemental materials.

The raw FASTQ-format data were subjected to quality control and evaluation, whereby low-quality or undetected sequences, sequences contaminated by splices, high-coverage repeats and short-read-length sequences were filtered out. High-quality sequencing data were compared with the human reference genome GRCh37 (hg19) using Bowtie2 v2.4.3 (29), enabling removal of human host sequences. The remaining sequences were aligned with the previously constructed reference database using Kraken2 v2.1.0 to annotate pathogen genomes and identify pathogens present in the sample. Genomic data on bacteria, fungi, viruses, parasites, and other pathogenic microorganisms were obtained from NCBI GenBank. These data were subjected to genomic filtering, screening, and validation to construct a reference database of pathogenic microorganisms that was suitable for clinical application. Pathogen positivity was determined using the normalized reads per 10 million (nRPTM), which was defined as the number of pathogen-specific reads per 10 million reads.

**Analytical validation of mNGS performance characteristics.** The matrix was a mixture of 100 mL ascites from five patients without infection. According to the ascites culture results reported in previous studies (9, 17, 18), two panels comprising six Gram-positive bacteria, five Gram-negative bacteria and one fungus were constructed to evaluate the analytical performance of mNGS testing (Table S1).

Briefly, limits of detection and linearity were obtained from a series of dilutions of 12 standard strains

within a 4-log range using probit and fitting analyses. Precision was determined using PCs and NCs over six consecutive sequencing runs for intraassay reproducibility and over five times conducted in parallel on the same run for interassay reproducibility. Interference was determined using related species in the same genus mixed in different proportions. Samples with PCs were held at a temperature of 4℃ for 0 h, 12 h, and 24 h for test stability analysis.

Ascitic samples of 205 patients were randomly divided into training ($n = 67$) and validation sets ($n = 138$). In the training set, the ROC curve was generated using nRPTM based on the traditional culture standard. The threshold for pathogen detection was established according to the optimal Youden index. Subsequently, accuracy was analyzed in the validation set based on the traditional culture and composite standard (culture, digital PCR, and clinical adjudication).

**Digital droplet PCR.** We used the digital droplet PCR (ddPCR) to further analyze the negative results detected by mNGS and culture. The assay was performed according to previously described methodology (Text S1) (30).

**Statistical analysis.** Data are presented as mean $\pm$ standard deviation or median with a range. The Wilcoxon-Mann-Whitney test was used for comparison between groups. The Kruskal–Wallis test was used for continuous variables, and Fisher's exact test was used for categorical variables. The Kappa test was used for consistency analysis, and the McNemar test was used to compare the diagnostic performance between the traditional culture and mNGS. At the pathogen level, 95% confidence intervals (CI) of the sensitivity and specificity were determined using Wilson's method (31). Statistical analyses were performed using SPSS software, version 24 (IBM, Armonk, NY), and Prism 8 (GraphPad, La Jolla, CA). In the two-sided test, $P$-values $<0.05$ were considered significant.

**Data availability.** Data generated or analyzed during this study are available from the corresponding author on reasonable request. Sequencing data that support the finding of this study (with human reads removed) have been deposited in NCBI SRA and can be accessed with the BioProject identifier PRJNA913604.

## SUPPLEMENTAL MATERIAL

Supplemental material is available online only.
**SUPPLEMENTAL FILE 1**, XLSX file, 0.02 MB.
**SUPPLEMENTAL FILE 2**, PDF file, 0.7 MB.

## ACKNOWLEDGMENTS

We thank Peizhi Li at Illumina China for providing comprehensive suggestions for this project and assistance with high-throughput sequencing.

This work was supported by the Beijing Municipal Science & Technology Commission, Administrative Commission of Zhongguancun Science Park (grant number Z211100002921004), and Clinical Research Special Fund of Wu Jieping Medical Foundation of China (grant number 320.6750.2022-08-1), the Capital Health Research and Development of Special Fund Program (grant number 2022-2-2183).

Hao-Xin Wu, Fei-Li Wei, Wei Zhang, Jie Han, Shan Guo, Zheng Wang, De-Xi Chen, Wei Hou, and Zhong-Jie Hu declare that they have no competing interests.

H.-X.W., F.-L.W., and W.H. conceived the study; F.-L.W. and Z.-J.H. contributed to the study design; H.-X.W. supervised all aspects of the study; W.H., W.Z., Z.W., and J.H. were responsible for clinical data collection and verification; D.-X.C. and S.G. were responsible for results collection and monitoring in the laboratory; H.-X.W. and F.-L.W. wrote the first draft of the manuscript. All authors critically reviewed the manuscript and contributed to writing-editing and approved the final version.

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
