## [Reviewer comments · Microbiology Spectrum]

Microbiology Spectrum

Clinical evaluation of metagenomic next-generation sequencing method for the diagnosis of suspected ascitic infection in patients with liver cirrhosis in a clinical laboratory

Hao-Xin Wu, Feili Wei, Wei Zhang, Jie Han, Shan Guo, Zheng Wang, Dexi Chen, Wei Hou, and Zhongjie Hu

Corresponding Author(s): Zhongjie Hu, Beijing YouAn Hospital

Review Timeline:

Submission Date:	July 29, 2022
Editorial Decision:	September 16, 2022
Revision Received:	October 20, 2022
Accepted:	November 21, 2022

Editor: Vittal Ponraj

Reviewer(s): The reviewers have opted to remain anonymous.

Transaction Report:

DOI: <https://doi.org/10.1128/spectrum.02946-22>

September 16, 2022

Dr. Zhong-Jie Hu
Beijing YouAn Hospital
Beijing
China

Re: Spectrum02946-22 (Clinical evaluation of metagenomic next-generation sequencing method for the diagnosis of suspected ascitic infection in patients with liver cirrhosis in a clinical laboratory)

Dear Dr. Zhong-Jie Hu:

Thank you for submitting your manuscript to Microbiology Spectrum. Your manuscript was peer-reviewed by two reviewers and their comments/queries are listed below for your consideration and response.

In addition to the reviewers comments, kindly address some of the minor comments included here:

Lines 110-111- Error message

Supplementary Figure 2b- Staphylococcus epidermidis repeated twice in the graph

Supplementary Line 157- DNA abstraction- Is that a Typo? (in-lieu DNA extraction)?

Link Not Available

While we are willing to consider a revised version of this paper at Spectrum, it would be in your best interest to improve the writing. I recommend that you ask a colleague of yours who is a native English speaker to read and provide you some feedback on the writing. You are also welcome to use one of the services here: <https://journals.asm.org/content/language-editing-services>
While we are willing to consider a revised version of this paper at Spectrum, it would be in your best interest to improve the writing. I recommend that you ask a colleague of yours who is a native English speaker to read and provide you some feedback on the writing. You are also welcome to use one of the services here: <https://journals.asm.org/content/language-editing-services>

Sincerely,

Vittal Prakash Ponraj Ph.D SM(ASCP)

Journals Department
Reviewer comments:

Reviewer #1 (Comments for the Author):

This is a well-constructed study with a thorough analysis of how mNGS compares with conventional culture for diagnosis of SBP. Adding digital droplet PCR and clinical adjudication as a secondary comparators given the sensitivity limitations of culture was an interesting approach. Comparison of the mNGS findings for SBP and bacterascites provides valuable information about the microbes associated with each condition. Certain conclusions are not clear and the manuscript would be strengthened with clarifications of these items. Other major and minor comments are outlined below.

Major:

1. In lines 180-182 the authors state WBC is increased in SBP as compared to bacterascites and no-AFI groups. Aren't SBP patients by definition supposed to have high WBC (>250). This is expected based on how patients are categorized.
2. Are there thoughts on why there was an increase in mortality with polymicrobial infections? The authors mention that Gram negative organisms are associated with the polymicrobial infections. Are there more details regarding this? Was it not possible to grow some of the Gram negative organisms because they are fastidious or because of pretreatment with antibiotics?
3. In lines 267-268 the authors state ddPCR had the advantage of detecting low levels of bacterial infection compared to culture and mNGS. Are the authors stating mNGS was not able to detect *S. pertenkovi*, *M. luteus*, and *S. pneumoniae* because of lower sensitivity as compared to ddPCR? If so, it might be worth providing analytical sensitivity and specificity data comparing ddPCR and mNGS.
4. In line 268 the authors mention that the organisms identified by ddPCR alone to be low pathogenicity. *S. pneumoniae* was one of the organisms identified and I would not consider this a low pathogenicity organism because it is a primary cause of SBP.
5. In lines 272-273 the authors state that mNGS detected cases faster than culture. Can the authors provide data on how much faster mNGS was with turnaround time information? This could be provided in supplementary material.
6. In lines 278-279 the authors mention that timely adjustment of antibiotics were made because of mNGS results. If antibiotics were adjusted because of mNGS results, can you provide information on how they were readjusted (e.g. narrowed, broadened, discontinued)? This could be included as supplementary material.
7. In line 306 the authors state that there is no performance evaluation of parasites due to the absence of standard strains. If this is the case, this should be listed as a limitation.
8. In lines 321-323 the authors state that concentration and characteristics of bactDNA detected by molecular technology and the entire clinical condition must be taken into account in considering the clinical significance. When referring to molecular technology is this referring to ddPCR or mNGS or both? If significance is determined by droplet PCR and expert review of each case than how does mNGS play a role?
9. In lines 329-330, the authors mention that antibiotic treatment is delayed for polymicrobial infections but do not provide an explanation. Why is this the case?
10. In lines 331-333 the authors state that patients with SBP caused by Gram-negative bacteria may be associated with mixed infections and that these patients have an increase in 6-month mortality. What are possible explanations for the increase in mortality for this group?
11. In lines 344-345 the authors state "only six patients had fungal infections in our study; therefore, it is necessary to develop specific tests for peritoneal fungal infection". Are the authors stating this because it is a limitation of mNGS to identify fungal infections or because this is a characteristic of peritoneal infections? This is not clear based on how the sentence is worded.
12. Would consider adding a section in the discussion describing the impact of mNGS on the case series of 8 prospective patients with peritonitis.

Minor:

1. When referring to Gram negative or Gram positive the "G" in Gram should be capitalized because it is a proper name.
2. In line 77 "lacked of accuracy analyses" could simply be stated "lacked accuracy".
3. In line 81 it would be clearer to use the wording "long turnaround time" rather than "long period".

4. In line 101 would substitute "diagnostic value" for "diagnosis value".

5. In line 327 the authors mention that patients with advanced liver cirrhosis have a mechanism for defense against bacterial invasion. What is this mechanism because this is not universally understood?

6. In lines 338-340 the authors describe their cooperation with an independent clinical laboratory for bioinformatics analysis as a limitation. This does not sound like a limitation. If this was a limitation the authors need to clearly state what the limitation of this collaboration was.

Reviewer #2 (Public repository details (Required)):

All the metagenomics data should be available

Reviewer #2 (Comments for the Author):

The paper from Wu et al entitled "Clinical evaluation of metagenomic next-generation sequencing method for the diagnosis of suspected ascitic infection in patients with liver cirrhosis in a clinical laboratory". This work evaluates the use of Metagenomic next-generation sequencing (mNGS) in the clinical settings. As a proof of concept, the authors performed a study on samples obtained from ascites. The results indicate that mNGS is faster, more accurate and allows for rapid treatment of infections. However, due to the cost's panels should be designed for specific pathologies.

In material and methods.

Line 131. "sequencing data were compared with the human reference genome GRCh37 (hg19) using alignment software"  Which alignment software?

Has the digital droplet PCR (ddPCR) method been validated previously? If the method you are using is not a commercially available, what quality controls have been used in the validation? Only the bacterial dilution? Is there an internal control for sample quality and PCR quality?

How were the samples inoculated in the microbiology laboratory? It is important to know that all the culturing was appropriate and that the negative results are not because the sample processing was not appropriate. There are not details at all about the microbiology analysis.

Results section

Patient characteristics. Could the authors explain in more detail what do they mean by "incomplete data on review" (line 182)? Do they mean that no clinical data was collected?

In general:

Attention to the reference, some of them appeared label as error instead a citation

Font size changes in some places along the text, please revise that is consistent all along the manuscript

Bacterial names should be italics. Along the manuscript there are some errors but on the tables none of them are in italics

In the figure 2 some of the figures are not clear as the writing goes on top of the figures, please correct that

Staff Comments:

Preparing Revision Guidelines

- Point-by-point responses to the issues raised by the reviewers in a file named "Response to Reviewers," NOT IN YOUR COVER LETTER.
- Upload a compare copy of the manuscript (without figures) as a "Marked-Up Manuscript" file.
- Each figure must be uploaded as a separate file, and any multipanel figures must be assembled into one file.
- Manuscript: A .DOC version of the revised manuscript

- Figures: Editable, high-resolution, individual figure files are required at revision, TIFF or EPS files are preferred

Please return the manuscript within 60 days; if you cannot complete the modification within this time period, please contact me. If you do not wish to modify the manuscript and prefer to submit it to another journal, please notify me of your decision immediately so that the manuscript may be formally withdrawn from consideration by Microbiology Spectrum.

This is a well-constructed study with a thorough analysis of how mNGS compares with conventional culture for diagnosis of SBP. Adding digital droplet PCR and clinical adjudication as a secondary comparators given the sensitivity limitations of culture was an interesting approach. Comparison of the mNGS findings for SBP and bacterascites provides valuable information about the microbes associated with each condition. Certain conclusions are not clear and the manuscript would be strengthened with clarifications of these items. Other major and minor comments are outlined below.

Major:

1. In lines 180-182 the authors state WBC is increased in SBP as compared to bacterascites and no-AFI groups. Aren't SBP patients by definition supposed to have high WBC (>250). This is expected based on how patients are categorized.
2. Are there thoughts on why there was an increase in mortality with polymicrobial infections? The authors mention that Gram negative organisms are associated with the polymicrobial infections. Are there more details regarding this? Was it not possible to grow some of the Gram negative organisms because they are fastidious or because of pretreatment with antibiotics?
3. In lines 267-268 the authors state ddPCR had the advantage of detecting low levels of bacterial infection compared to culture and mNGS. Are the authors stating mNGS was not able to detect *S. pertenkovi*, *M. luteus*, and *S. pneumoniae* because of lower sensitivity as compared to ddPCR? If so, it might be worth providing analytical sensitivity and specificity data comparing ddPCR and mNGS.
4. In line 268 the authors mention that the organisms identified by ddPCR alone to be low pathogenicity. *S. pneumoniae* was one of the organisms identified and I would not consider this a low pathogenicity organism because it is a primary cause of SBP.
5. In lines 272-273 the authors state that mNGS detected cases faster than culture. Can the authors provide data on how much faster mNGS was with turnaround time information? This could be provided in supplementary material.
6. In lines 278-279 the authors mention that timely adjustment of antibiotics were made because of mNGS results. If antibiotics were adjusted because of mNGS results, can you provide information on how they were readjusted (e.g. narrowed, broadened, discontinued)? This could be included as supplementary material.
7. In line 306 the authors state that there is no performance evaluation of parasites due to the absence of standard strains. If this is the case, this should be listed as a limitation.
8. In lines 321-323 the authors state that concentration and characteristics of bactDNA detected by molecular technology and the entire clinical condition must be taken into account in considering the clinical significance. When referring to molecular technology is this referring to ddPCR or mNGS or both? If significance is determined by droplet PCR and expert review of each case than how does mNGS play a role?

9. In lines 329-330, the authors mention that antibiotic treatment is delayed for polymicrobial infections but do not provide an explanation. Why is this the case?
10. In lines 331-333 the authors state that patients with SBP caused by Gram-negative bacteria may be associated with mixed infections and that these patients have an increase in 6-month mortality. What are possible explanations for the increase in mortality for this group?
11. In lines 344-345 the authors state “only six patients had fungal infections in our study; therefore, it is necessary to develop specific tests for peritoneal fungal infection”. Are the authors stating this because it is a limitation of mNGS to identify fungal infections or because this is a characteristic of peritoneal infections? This is not clear based on how the sentence is worded.
12. Would consider adding a section in the discussion describing the impact of mNGS on the case series of 8 prospective patients with peritonitis.

Minor:

1. When referring to Gram negative or Gram positive the “G” in Gram should be capitalized because it is a proper name.
2. In line 77 “lacked of accuracy analyses” could simply be stated “lacked accuracy”.
3. In line 81 it would be clearer to use the wording “long turnaround time” rather than “long period”.
4. In line 101 would substitute “diagnostic value” for “diagnosis value”.
5. In line 327 the authors mention that patients with advanced liver cirrhosis have a mechanism for defense against bacterial invasion. What is this mechanism because this is not universally understood?
6. In lines 338-340 the authors describe their cooperation with an independent clinical laboratory for bioinformatics analysis as a limitation. This does not sound like a limitation. If this was a limitation the authors need to clearly state what the limitation of this collaboration was.

Dear Editor and Reviewers:

Thank you for your letter and the reviewer's comments concerning our manuscript "Clinical evaluation of metagenomic next-generation sequencing method for the diagnosis of suspected ascitic infection in patients with liver cirrhosis in a clinical laboratory" (ID: Spectrum02946-22). Those comments are constructive for us to revise and improve our paper. We have studied these comments carefully and tried our best to change and improve the manuscript. Modified portions are marked in red on the paper. The leading corrections in the form and the responses to the reviewer's comments are as follows:

Respond to reviewers' comments

1. In lines 180-182 the authors state WBC is increased in SBP as compared to bacterascites and no-AFI groups. Aren't SBP patients by definition supposed to have high WBC (>250). This is expected based on how patients are categorized.

Response : The diagnosis of SBP is based on neutrophil count in ascitic fluid (PMN) of >250/mm³. From the demographic data, the SBP cohort showed increased white blood cell and neutrophil counts in blood and in ascites compared with those in bacterascites and no-AFI groups (P<0.001).

We have made changes on lines 192-194 and marked.

2. Are there thoughts on why there was an increase in mortality with polymicrobial infections? The authors mention that Gram negative organisms are associated with the polymicrobial infections. Are there more details regarding this? Was it not possible to grow some of the Gram negative organisms because they are fastidious or because of

pretreatment with antibiotics?

Response : Thanks very much for your comments and detailed explanation.

The increased mortality of polymicrobial infections may be related to the following reasons: 1) selected the incorrect antibiotics due to polymicrobial infection leads to a delay in treatment in clinical practice; 2) cirrhosis-associated immune dysfunction, which contributes to an increased risk of infections and mortality (1); 3) previous research has shown that polymicrobial infection is “polymicrobial biofilm formation”, in which polymicrobial interactions of different pathogens lead to an increased tolerance to antimicrobial agents and enhanced polymicrobial biomass (2). Our study found an increased mortality with polymicrobial infections, and similar studies have been reported in recent years (3, 4, 5).

In fact, Gram-negative organisms are associated with the polymicrobial infections. There are several possible reasons for this. Firstly, more recently, there has been a shift toward Gram-positive organisms, particularly in nosocomial SBP (6). Blood culture is limited by growth inhibition by prior use of antibiotics. Secondly, the possible mechanism suggests that Gram-negative bacteria are helpful for forming the bacterial biofilm matrices that might act as scaffolds that facilitate the attachment of certain bacteria (7). However, the mechanism behind this infection remains unknown and requires further research.

We have added this part of the explanation to the Discussion of line 339-350.

3. In lines 267-268 the authors state ddPCR had the advantage of detecting low levels of bacterial infection compared to culture and mNGS. Are the authors stating mNGS was not able to detect *S. pertenkovi*, *M. luteus*, and *S. pneumoniae* because of lower sensitivity as compared to ddPCR? If so, it might be worth providing analytical

sensitivity and specificity data comparing ddPCR and mNGS.

Response : I am sorry that I didn't make myself clear. In our opinion, mNGS was able to detect *S. pertenkovi*, *M. luteus*, and *S. pneumoniae*, whereas ddPCR could not detect those pathogens. However,

We compared the diagnostic accuracy rate between culture, mNGS, and ddPCR methods in diagnosing SBP (Figure 1A,B). The results showed that pathogens were detected in 50.0% (33/66) of patients by culture and in 54.5% (36/66) of patients by mNGS and in 69.7% (46/66) by ddPCR. In addition, the specificity and PPA of ddPCR for suspected SBP were higher than that of culture and mNGS (87.1% vs 71.9% and 67.6%, 71.9% vs 45.8% and 44.5%, $P < 0.05$), indicating that ddPCR may have more advantages in diagnosing abdominal infection compared with culture and mNGS (Figure 1C).

Considering the reviewer's comments, we made the complement and changes in line 275-286 of manuscript and supplementary materials.

4. In line 268 the authors mention that the organisms identified by ddPCR alone to be low pathogenicity. *S. pneumoniae* was one of the organisms identified and I would not consider this a low pathogenicity organism because it is a primary cause of SBP.

Response : Thank you for pointing out a meaningful question.

S. pneumoniae was one of the pathogenic organisms identified, especially in community-acquired pleural infection (7). However, the main organisms commonly appearing in peritonitis included *E. coli*, *E. faecalis*, *S. aureus*, *K. pneumoniae* and among others, whereas *S. pneumoniae* is rare in peritoneal infections (8,9,10). Our study showed that in 83 patients with organisms detected by mNGS, only three patients (4%) were complicated with *S.*

pneumoniae infection, none of which detected by culture. Among them, one SBP patient mixed with infection of *S. aureus* was characterized by fever, elevated PMN and C-reactive protein, and was treated with teicoplanin antibiotic. Two patients did not appear clinical symptoms and were not treated, and both of them were relieved of abdominal distension and discharged.

5. In lines 272-273 the authors state that mNGS detected cases faster than culture. Can the authors provide data on how much faster mNGS was with turnaround time information? This could be provided in supplementary material.

Response : The workflow of mNGS testing included DNA extraction, library preparation, sequencing, and data analysis, which rapidly completed within 24 h. The turnaround time were approximately 2 h, 3 h, and 11 h, respectively. Detailed information was described in Figure 2 and added in supplementary material.

6. In lines 278-279 the authors mention that timely adjustment of antibiotics were made because of mNGS results. If antibiotics were adjusted because of mNGS results, can you provide information on how they were readjusted (e.g. narrowed, broadened, discontinued)? This could be included as supplementary material.

Response : Thanks for your comments. Briefly, five patients were culture-negative but mNGS-positive, antibiotics were given according to mNGS results. Two patients performed incorrect antibiotic prophylaxis use and were adjusted to appropriate antibiotic therapy according to mNGS results. One patient mixed with fungal infection and were added anti-fungal drugs for further treatment.

The detailed information of 8 patients has been included in supplementary materials.

7. In line 306 the authors state that there is no performance evaluation of parasites due to the absence of standard strains. If this is the case, this should be listed as a limitation.

Response : Thanks for your suggestions. The performance evaluation of parasites had been listed as a limitation in line 390-393.

8. In lines 321-323 the authors state that concentration and characteristics of bactDNA detected by molecular technology and the entire clinical condition must be taken into account in considering the clinical significance. When referring to molecular technology is this referring to ddPCR or mNGS or both? If significance is determined by droplet PCR and expert review of each case than how does mNGS play a role?

Response : One of the limitations of molecular diagnostic technology is high false-positive rate, and the frequent detection of contaminants and colonizing pathogens both affects the specificity of NGS and ddPCR in diagnosing peritoneal infection. In our study, we mainly introduced the detected accuracy rate of pathogens and diagnostic value of infection based on mNGS testing.

However, it is essential to detect bactDNA levels based on ddPCR, and the clinical applications of these two methods play vital roles in different clinical scenes. Our results showed that mNGS detected a broad range of pathogens, whereas ddPCR was more rapid (4h vs. 24h) and more sensitive than mNGS in diagnosing SBP, which was consistent with recent studies (11,12). In addition, the ddPCR showed a great potential to identify and exclude the common peritoneal pathogens, whereas mNGS is more appropriate in the diagnosis of rare infections and intractable diseases.

Considering the reviewer's comments, we made the complement and changes in line

362-364.

9. In lines 329-330, the authors mention that antibiotic treatment is delayed for polymicrobial infections but do not provide an explanation. Why is this the case?

Response : We made an explanation about this sentence. When the results of culture showed monomicrobial infections but mNGS showed polymicrobial infections, effective antibiotic coverage of pathogens could be limited merely based on culture, leading to severe clinical outcomes. We have made changes in line 339-341 of manuscript.

10. In lines 331-333 the authors state that patients with SBP caused by Gram-negative bacteria may be associated with mixed infections and that these patients have an increase in 6-month mortality. What are possible explanations for the increase in mortality for this group?

Response : Thanks very much for your comments.

As mentioned above, we found that patients with SBP caused by Gram-negative bacteria may be associated with mixed infections and patients with polymicrobial infection had an increase in mortality. The possible explanations for the increase in mortality are as follows: on the one hand, from polymicrobial interaction, the process of bacterial co-aggregation enhanced the pathogenesis and evolution of peritoneal infection. On the other hand, from host immune system, liver-cirrhosis immune dysfunction may predispose the host to more severe infection by translocated bacteria. In addition, incorrect antibiotic prophylaxis use contributed to a delay in treatment.

11. In lines 344-345 the authors state "only six patients had fungal infections in our study; therefore, it is necessary to develop specific tests for peritoneal fungal infection".

Are the authors stating this because it is a limitation of mNGS to identify fungal infections or because this is a characteristic of peritoneal infections? This is not clear based on how the sentence is worded.

Response : I am sorry that I didn't make myself clear.

EASL clinical guideline suppose that spontaneous fungal peritonitis is a rare, less recognized and studied complication, occurring in <5% of cases, but observational data suggest a worse prognosis (13,14,15). We did not verify the fungal infection because only 6 patients with fungal infection were detected by mNGS in our study. Fungal infections are also the characteristic of peritoneal infections, so it is necessary to develop a specific tests for peritoneal fungal infection.

We made the correction and marked in manuscript.

12. Would consider adding a section in the discussion describing the impact of mNGS on the case series of 8 prospective patients with peritonitis.

Response : We are very grateful for your comments and suggestions.

Through the case series of 8 prospective patients with peritonitis, we found that mNGS testing not only provides a solid basis for clinical precision medicine, but also performs better antibiotic management and further improves clinical outcomes. In addition, the average length of patient stay is 12-15 days. Recent studies have also shown that rapid and accurate identification of pathogens using nucleic acid tests decreases healthcare costs as well as mortality (16,17). However, the mNGS method is still relatively expensive compared with traditional culture.

we made the complement and changes in line 384-389.

13. In line 327 the authors mention that patients with advanced liver cirrhosis have a mechanism for defense against bacterial invasion. What is this mechanism because this is not universally understood?

Response : The mechanism includes two lines for defense against bacterial invasion. The gastrointestinal barrier and gut associated lymphoid tissue immune response constitute the first line of defense, and peritoneal immune function is the crucial second line of defense to prevent SBP (2,18).

14. When referring to Gram negative or Gram positive the "G" in Gram should be capitalized because it is a proper name.

In line 77 "lacked of accuracy analyses" could simply be stated "lacked accuracy".

In line 81 it would be clearer to use the wording "long turnaround time" rather than "long period".

In line 101 would substitute "diagnostic value" for "diagnosis value".

In lines 338-340 the authors describe their cooperation with an independent clinical laboratory for bioinformatics analysis as a limitation. This does not sound like a limitation. If this was a limitation the authors need to clearly state what the limitation of this collaboration was.

Response : Thanks very much for your comments and detailed observation.

We have made changes and marked respectively in this manuscript.

Respond to reviewers' comments

1. Line 131. "sequencing data were compared with the human reference genome GRCh37 (hg19) using alignment software"  Which alignment software?

Response : High-quality sequencing data were compared with the human reference genome GRCh37 (hg19) using alignment software Bowtie2 v2.4.3 (19), enabling removal of human host sequences.

2. Has the digital droplet PCR (ddPCR) method been validated previously? If the method you are using is not a commercially available, what quality controls have been used in the validation? Only the bacterial dilution? Is there an internal control for sample quality and PCR quality?

Response : Thanks for your comments. The ddPCR method was constructed based on in-house protocols, and the analytical performances were performed and validated using in-vitro diagnostic reagents (370026-201801, National Institutes for Food and Drug Control). Our results showed that the assay had great specificity, stability, interference, linearity of 0.97-0.99 of R squared, and limits of detection of 20-45 copies/ μ l. This study was published in the journal *Frontiers in Cellular and Infection Microbiology* (20).

In order to perform quality controls, we detected human NAKG gene of every sample for internal control, and the assay was quality controlled using negative external controls (DNase/RNase-free water) and positive external controls (known quantities of DNA from organisms) included in every batch.

3. How were the samples inoculated in the microbiology laboratory? It is important to know that all the culturing was appropriate and that the negative results are not because the sample processing was not appropriate. There are not details at all about the microbiology analysis.

Response : We are grateful for your comments.

We process the samples in the microbiology laboratory as follows: Ascitic fluid obtained from each patient was injected into automatic blood culture bottles (Aerobic, Anaerobic and Myco/F Lytic bottles, 8–10 mL per bottle) to culture aerobic bacteria, anaerobic bacteria, mycobacteria and fungus in BD BACTEC FX and BD BACTECTM 9120 Automated blood culture system. When the system showed a positive growing signal, the sample was extracted from the bottle for Gram staining or Acid-fast staining to make smear microscopy, followed by subculture on Columbia blood agar and Maconkey Aga plate at 37°C with 5% CO₂ for aerobic bacteria or without oxygen for anaerobic bacteria. If the system showed a positive blood culture signal for Myco/F Lytic bottle, besides smear microscopy, the sample was extracted from the bottle and cultured on SDA agar for fungus at 28°C. The pathogens cultured on the agar were further identified by matrix-assisted laser desorption-ionization time-of-flight mass spectrometry (MALDI-TOF MS; VITEK MS system, bioMérieux, France). The negative result without positive growing signal was reported when the samples were cultured 5 days for aerobic and anaerobic bacteria, 14 days for fungus, 42 days for mycobacteria in the automatic blood culture system.

The detailed information has been included in supplementary methods.

4. Could the authors explain in more detail what do they mean by "incomplete data on review" (line 182)? Do they mean that no clinical data was collected?

Response : In our study, the clinical or laboratory data in four patients were incomplete, which included one patient with sequencing failure and three patients with missing data such as blood cell count, C-reactive protein and procalcitonin level, and among others. We have made changes in manuscript of line 102-103.

5. Font size changes in some places along the text, please revise that is consistent all along the manuscript

Bacterial names should be italics. Along the manuscript there are some errors but on the tables none of them are in italics

In the figure 2 some of the figures are not clear as the writing goes on top of the figures, please correct that

Response : Thanks very much for your comments and detailed observation.

We have made changes and marked respectively in this manuscript.

Reference:

1. Albillos A, Martin-Mateos R, Van der Merwe S, Wiest R, Jalan R, Álvarez-Mon M. 2022. Cirrhosis-associated immune dysfunction[J]. *Nat Rev Gastroenterol Hepatol*; 19:112-134.
2. Peters BM, Jabra-Rizk MA, O'May GA, Costerton JW, Shirtliff ME. 2012. Polymicrobial Interactions: Impact on Pathogenesis and Human Disease[J]. *Clin Microbiol Rev*; 25:193-213.
3. Jiao M, Ma X, Li Y, Wang H, Liu Y, Guo W, Lv J. 2022. Metagenomic next-generation sequencing provides prognostic warning by identifying mixed infections in nocardiosis[J]. *Front Cell Infect Microbiol*. 31;12:894678.
4. Hardak E, Avivi I, Berkun L, Raz-Pasteur A, Lavi N, Geffen Y, et al. 2016. Polymicrobial pulmonary infection in patients with hematological malignancies: prevalence, co-pathogens, course and outcome. *Infection*;44:491–7.
5. Yo CH, Hsein YC, Wu YL, Hsu WT, Ma MH, Tsai CH, et al. 2019. Clinical predictors

- and outcome impact of community-onset polymicrobial bloodstream infection. *Int J Antimicrob Agents*. 54:716-722.
6. Biggins SW, Angeli P, Garcia-Tsao G, Ginès P, Ling SC, Nadim MK, Wong F, Kim WR. 2021. Diagnosis, Evaluation, and Management of Ascites, Spontaneous Bacterial Peritonitis and Hepatorenal Syndrome: 2021 Practice Guidance by the American Association for the Study of Liver Diseases[J]. *Hepatology*;74:1014-1048.
 7. Kanellakis NI, Wrightson JM, Gerry S, Ilott N, Corcoran JP, Bedawi EO, et al. 2022. The bacteriology of pleural infection (TORPIDS): an exploratory metagenomics analysis through next generation sequencing[J]. *Lancet Microbe*. 3(4):e294-e302.
 8. Shi L, Wu D, Wei L, Liu S, Zhao P, Tu B, Xie Y, Liu Y, Wang X, Liu L, Zhang X, Xu Z, Wang F, Qin E. 2017. Nosocomial and Community-Acquired Spontaneous Bacterial Peritonitis in patients with liver cirrhosis in China: Comparative Microbiology and Therapeutic Implications[J]. *Sci Rep*;7:46025.
 9. Hardick J, Won H, Jeng K, Hsieh YH, Gaydos CA, Rothman RE, Yang S. 2012. Identification of bacterial pathogens in ascitic fluids from patients with suspected spontaneous bacterial peritonitis by use of broad-range PCR (16S PCR) coupled with high-resolution melt analysis[J]. *J Clin Microbiol*;50(7):2428-32.
 10. Kim T, Hong SI, Park SY, Jung J, Chong YP, Kim SH, Lee SO, Kim YS, Woo JH, Lim YS, Sung H, Kim MN, Choi SH. 2016. Clinical Features and Outcomes of Spontaneous Bacterial Peritonitis Caused by *Streptococcus pneumoniae*: A Matched Case-Control Study[J]. *Medicine (Baltimore)*;95(22):e3796.
 11. Hu B, Tao Y, Shao Z, Zheng Y, Zhang R, Yang X, Liu J, Li X, Sun R. 2022. A

- Comparison of Blood Pathogen Detection Among Droplet Digital PCR, Metagenomic NextGeneration Sequencing, and Blood Culture in Critically Ill Patients With Suspected Bloodstream Infections[J]. *Front Microbiol.* 12:641202.
12. Wu J, Tang B, Qiu Y, Tan R, Liu J, Xia J, Zhang J, Huang J, Qu J, Sun J, Wang X, Qu H. 2022. Clinical validation of a multiplex droplet digital PCR for diagnosing suspected bloodstream infections in ICU practice: a promising diagnostic tool[J]. *Crit Care.* 26(1):243.
 13. European Association for the Study of the Liver. 2018. EASL Clinical Practice Guidelines for the management of patients with decompensated cirrhosis[J]. *J Hepatol*;69:406-460.
 14. Gravito-Soares M, Gravito-Soares E, Lopes S, Ribeiro G, Figueiredo P. 2017. Spontaneous fungal peritonitis: a rare but severe complication of liver Cirrhosis[J]. *Eur J Gastroenterol Hepatol*;29:1010–1016
 15. Li B, Yang C, Qian Z, Huang Y, Wang X, Zhong G, Chen J. 2021. Spontaneous Fungal Ascites Infection in Patients with Cirrhosis: An Analysis of 10 Cases[J]. *Infect Dis Ther*; 10(2):1033-1043.
 16. Piantadosi A, Mukerji SS, Ye S, Leone MJ, Freimark LM, Park D, Adams G, Lemieux J, Kanjilal S, Solomon IH, Ahmed AA, Goldstein R, Ganesh V, Ostrem B, Cummins KC, Thon JM, Kinsella CM, Rosenberg E, Frosch MP, Goldberg MB, Cho TA, Sabeti P. 2021. Enhanced Virus Detection and Metagenomic Sequencing in Patients with Meningitis and Encephalitis[J]. *mBio*;12: e0114321.
 17. Perez KK, Olsen RJ, Musick WL, Cernoch PL, Davis JR, Land GA, Peterson LE,

- Musser JM. 2013. Integrating rapid pathogen identification and antimicrobial stewardship significantly decreases hospital costs[J]. *Arch. Pathol. Lab. Med.* 137, 1247–1254.
18. Stengel S, Quickert S, Lutz P, Ibidapo-Obe O, Steube A, Köse-Vogel N, Yarbakt M, Reuken PA, Busch M, Brandt A, Bergheim I, Deshmukh SD, Stallmach A, Bruns T. 2020. Peritoneal Level of CD206 Associates With Mortality and an Inflammatory Macrophage Phenotype in Patients With Decompensated Cirrhosis and Spontaneous Bacterial Peritonitis[J]. *Gastroenterology* 158(6):1745-1761.
 19. Langmead B, Salzberg SL. Fast gapped-read alignment with Bowtie 2. *Nat Methods* 2012;9:357–9.
 20. Wu HX, Hou W, Zhang W, Wang Z, Guo S, Chen DX, Li Z, Wei F, Hu Z. 2022. Clinical evaluation of bacterial DNA using an improved droplet digital PCR for spontaneous bacterial peritonitis diagnosis[J]. *Front Cell Infect Microbiol*;12:876495.

Figure 1. Analytical diagnostic accuracy rate of three detection methods.

A.

B.

C.

SBP	culture		mNGS		ddPCR	
	pos	neg	pos	neg	pos	neg
pos	33	33	36	30	46	20
neg	39	100	45	94	18	121

Sensitivity = 50.0%
Specificity = 71.9%

Sensitivity = 54.5%
Specificity = 67.6%

Sensitivity = 69.7%
Specificity = 87.1%

D.

Figure 2. The turnaround time information of mNGS.

November 21, 2022

Dr. Zhongjie Hu
Beijing YouAn Hospital
No.8, West Tou Tiao Community, Youan Men Wai Street, Fengtai District, Beijing
beijing
China

Re: Spectrum02946-22R1 (Clinical evaluation of metagenomic next-generation sequencing method for the diagnosis of suspected ascitic infection in patients with liver cirrhosis in a clinical laboratory)

Dear Dr. Zhongjie Hu:

Thank you for submitting the revised version of the manuscript that the previous reviewers seem to be satisfied with. Please make sure you incorporate the comments you provided in your response letter to appropriate sections of the manuscript, as deemed appropriate.

Your manuscript has been accepted, and I am forwarding it to the ASM Journals Department for publication. You will be notified when your proofs are ready to be viewed.

Sincerely,

Vittal Prakash Ponraj Ph.D., SM(ASCP)CM
Editor, Microbiology Spectrum

Journals Department
Supplemental Material: Accept
Supplemental file 1: Accept